# Cryo-EM structure and polymorphic maturation of a viral transduction enhancing amyloid fibril

Thomas Heerde [1] ✉, Desiree Schütz[2], Yu-Jie Lin[1], Jan Münch [2], Matthias Schmidt [1] & Marcus Fändrich [1]

Amyloid fibrils have emerged as innovative tools to enhance the transduction efficiency of retroviral vectors in gene therapy strategies. In this study, we used cryo-electron microscopy to analyze the structure of a biotechnologically engineered peptide fibril that enhances retroviral infectivity. Our findings show that the peptide undergoes a time-dependent morphological maturation into polymorphic amyloid fibril structures. The fibrils consist of mated cross-β sheets that interact by the hydrophobic residues of the amphipathic fibril-forming peptide. The now available structural data help to explain the mechanism of retroviral infectivity enhancement, provide insights into the molecular plasticity of amyloid structures and illuminate the thermodynamic basis of their morphological maturation.

Amyloid fibrils are fibrillar polypeptide aggregates with a cross-β structure[1]. Their formation inside the body is commonly associated with debilitating conditions such as neurodegenerative diseases or systemic amyloidosis[2]. Yet, there is increasing interest to exploit the structural scaffold of amyloid fibrils for different biotechnological purposes. Examples of biotechnologically exploited amyloid fibril structures are conductive nanowires[3], materials for the selective capture of carbon dioxide[4], catalytic agents[5], or enhancers of retroviral transduction[6]. The ability to enhance transduction rates of retroviral vectors may be used, for example, in gene therapeutic settings, such as in the chimeric antigen receptor T-cell therapy, which involves the use of viral vectors to genetically engineer patient's T-cells in order to target and kill cancer cells[7].

The latter applications originated from observations that amyloid fibrils formed from peptide fragments of human prostate acidic phosphatase drastically enhance the infectivity of human immunodeficiency virus (HIV)−1[8,9]. Similar activities were then associated with a range of other amyloid fibril systems[6,10,11], including the fibrils prepared from a 12-mer peptide derived from the HIV envelope protein gp120 and termed enhancing factor C (EF-C)[12]. EF-C exhibits one of the strongest enhancement effects on retroviral gene transfer and has been marketed under the brand name Protransduzin[13].

The biotechnological applications of amyloid fibrils are generally based on their unique biochemical properties that include a high chemical and physical stability, a modular assembly from short polypeptide chains and an extremely repetitive surface texture[2,14]. In case of the virus-enhancing activity it is thought that it arises from their ability of these fibrils to bind viral particles and to promote the interactions of the bound virus with the surface of human cells[10,15,16]. These features depend in turn on the overall charge of the fibrils[15,17], the length of the fibril forming peptide, the presence of amphiphilic residue patterns or the involvement of different charged amino acids[17]. In the absence of an experimentally determined structure of a virus-binding fibril, however, it is difficult to estimate how these peptide properties manifest themselves at the quaternary structural level of the assembled amyloid fibril.

To address this issue, we have used cryo-electron microscopy (cryo-EM) to determine the molecular structure of an amyloid fibril that is formed from peptide PNF-18. This 7-mer peptide (sequence CKFKFQF) was artificially designed as retroviral transduction enhancer and increases HIV-1 infection rates of TZM-bl cells by a factor of more than a hundred-fold[17]. Our findings show that the structure is based on a fundamental building principle. Each protofilament (PF) is built by two paired cross-β sheets that interact by the hydrophobic side of the amphipathic peptide, while the polar residues are pointing outwards,

[1]Institute of Protein Biochemistry, Ulm University, 89081 Ulm, Germany. [2]Institute of Molecular Virology, Ulm University Medical Center, 89081 Ulm, Germany. ✉ e-mail: thomas.heerde@uni-ulm.de

creating a positive surface. These structural features provide insights into the molecular basis of amyloid polymorphism and virus binding, and improves our understanding of the molecular basis of peptide nanofibrils as retroviral transduction enhancers.

## Results

### The incubation time affects the PNF-18 fibril morphology

Chemically synthesized PNF-18 peptide was dissolved at 0.3 mg/mL concentration in 50 mM HEPES buffer, pH 7, and incubated at room temperature for various periods of time (1 to 60 days). Aliquots were withdrawn from the reaction and analyzed with transmission electron microscopy (TEM). All samples contained large quantities of TEM-visible amyloid fibrils, but there were marked, time-dependent changes in the fibril morphology. Peptide solutions aged for one day contained relatively thin fibril morphologies with a width of around 5 nm (Fig. 1). These fibril morphologies progressively disappeared upon further incubation of the sample and thicker fibril structures became more prominent (Fig. 1). Associated with these changes was an increase in the heterogeneity of the fibril ensemble compared with the structures seen after 1 day of incubation (Fig. 1b). The aged fibrils showed well-resolved cross-overs, arising from their twisted structure.

### Quantitative analysis of the sample polymorphism

Analysis of PNF-18 fibril samples with cryo-EM confirmed this morphological evolution of the fibril structures, and identified three main fibril morphologies, termed here morphologies I, II and III in the order of their ascending width and cross-over distance (Fig. 2a, b). The width and crossover distances of these fibrils were $4.3 \pm 0.3$ nm and $15.9 \pm 1.2$ nm for Morphology I, $6.8 \pm 0.3$ nm and $31.6 \pm 1.9$ nm for Morphology II, as well as $9.4 \pm 0.7$ nm and $43.5 \pm 2.3$ nm for Morphology III ($n = 30$ each). The relative abundance of each morphology was determined based on the cross-over distance and the width of 500 randomly chosen fibrils from each sample, showing that the three morphologies accounted for 99% of the fibrils in the sample aged for 1 day (Fig. 2c). But similar to our observations made with negatively stained samples (Fig. 1), we found that increasing the incubation period to 14 days changed the morphological composition and reduced the percentage of Morphology I structures from 63% on day 1 to 6% after 2 weeks. The loss of Morphology I occurred was associated with

an increase of Morphologies II and III, which raised in abundance from 31% and 6% on day 1 to 53% and 31% on day 14. The remaining fibril morphologies, which showed even larger crossover distances and fibril widths (Fig. 2a, b), also increased within this time frame (<1% after 1 day and 10% after 14 days, Fig. 2c).

### The time-dependent morphological changes have little effect on infectivity enhancement

To confirm that the formed fibrils are able to enhance HIV-1 infectivity we added the fibrils to HIV-1 particles and TZM-bl cells. The fibrils showed, within the low-concentration range, a dose-dependent enhancement of viral infectivity, but this effect leveled off if the peptide concentration was raised to 50 μg/mL or more (Supplementary Fig. 1). A similar saturation of the infectivity enhancement at high fibril concentrations has been reported for other virus enhancing fibrils as well[6,12] and may arise from an entanglement of the fibrils into a large-sized meshwork that causes a breakdown of the strict dose-dependency. The infectivity enhancement of PNF-18 fibrils occurs in a similar range as that of EF-C fibrils, suggesting that the 7-mer peptide PNF-18 suffices to produce a similarly potent increase in infectivity. The infectivity enhancement did not depend substantially on the age of the tested fibril solution (Supplementary Fig. 1) and contrasts to the profound effects of the incubation time on the morphological composition of the sample (Fig. 1). We conclude that the specific morphology of PNF-18 fibrils is not crucial for their virus enhancing infectivity.

### Cryo-EM structure of Morphology II

In the next step, we subjected the cryo-EM data collected with Morphology II to methods with which the fibril 3D map could be reconstructed. We achieved a spatial resolution of 2.86 Å, based on the 0.143 Fourier shell correlation (FSC) criterion (Supplementary Table 1, Fig. 3a and Supplementary Fig. 2a). The 3D map was interpreted with a molecular model at a model resolution of 2.9 Å (Fig. 3b, Supplementary Fig. 2b and Supplementary Table 2). Projections of the model density corresponded well to the two-dimensional class averages as well as their power spectra (Supplementary Fig. 2c). The highest local resolution was in center of the fibril structure and the resolution declined towards the periphery of the cross-section (Supplementary

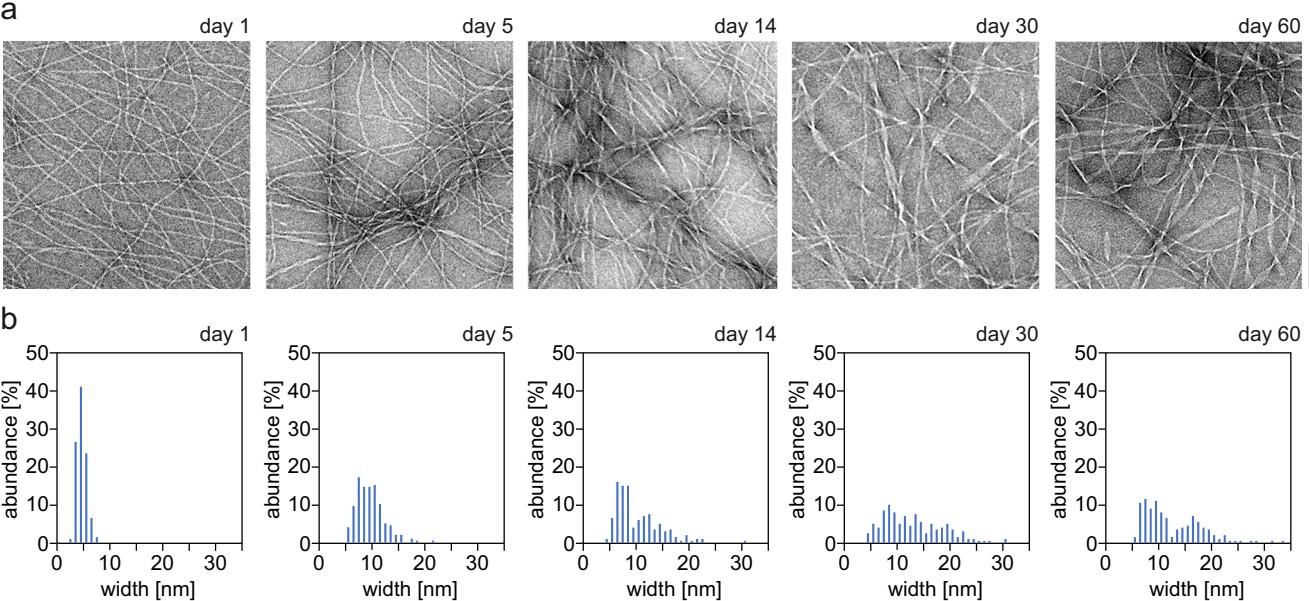

**Fig. 1 | Dependence of the fibril width on the incubation time. a** Negative Stain TEM images of in vitro formed PNF-18 fibrils at different times of incubation. Scalebar 100 nm. **b** Relative abundance of fibril widths of in vitro formed PNF-18 fibrils at different times of incubation.

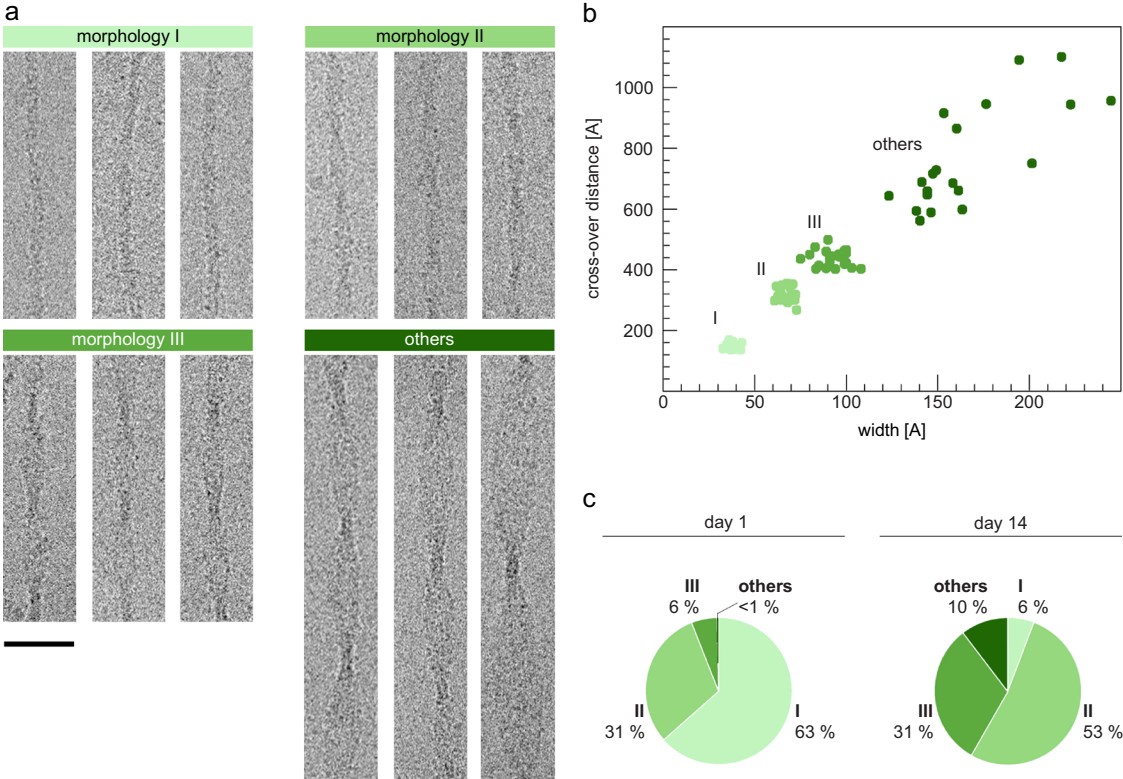

**Fig. 2 | Polymorphism of in vitro PNF-18 fibrils. a** Cryo-EM images of fibril morphologies I − III and others. Scalebar: 30 nm. **b** Crossover distance and fibril width values of fibril morphologies I − III and others ($n = 20$). **c** Relative abundance of fibril morphologies I − III and others in samples after one and 14 days of incubation ($n = 500$, per sample).

Fig. 3) The fibril consisted of six peptide stacks with cross-β sheet conformation that presented uniformly parallel strand-strand interactions in the direction of the fibril z-axis (Fig. 3c). There were, depending on the position of the peptide stack in the fibril structure, three topological types of peptide stacks (α, β and γ, Fig. 3b, c) that were arranged into pairs, which we refer to hereafter as PFs. Morphology II consists of three PFs, two peripheral ones and a central one, and six peptides in each molecular layer (Fig. 3b). The fibril shows C2 symmetry, consistent with the obtained 2D class averages (Fig. 3a, c). The fibril structure is polar and differs structurally at its two ends.

### Interactions of the peptides in the fibril

The peptide conformation is kinked and shows two chemically different sides: a hydrophobic side that is formed by residues Phe3, Phe5, and Phe7 and a polar one that is formed by the side chains of Cys1, Lys2, Lys4, Gln6 as well as the N- and C-terminal groups of the polypeptide backbone (Fig. 4a). The peptide's amphipathic properties lead to two modes by which two peptides (or cross-β sheets) may interact within the plane of the fibril cross-section: a hydrophobic and a polar contact mode. The hydrophobic contact mode occurs between the two type α peptide stacks and between a type β and a type γ stack (Fig. 4b). It involves the packing between relatively smooth surfaces that are formed by phenylalanine residues. The polar contact mode, which occurs between type α and β peptide stacks, involves an interdigitation of the side chains of residues Lys4 and Gln6 (Fig. 4c), similar to a steric zipper[18]. The zipper-like region is flanked on both sides by an intermolecular charge pair that is formed by the ε-amino group of Lys4 and α-carboxy group of the peptide C-terminus (Fig. 4c). Both contacts modes contain a face-to-face packing of the peptides in the plane of the cross-section.

The hydrophobic contact mode defines the PF core, while the polar contact mode occurs between two juxtaposed PFs (Fig. 5a). It significantly restrains, due to the interdigitation of the residues in

the zipper, the possible orientation of the peptide molecules in stacks α and β. Therefore, all peptide molecules connected through these interactions run almost perpendicular to the main fibril axis. Only the peptides in stack γ, which are bound through the hydrophobic contact mode to stack β, adopt an angle of almost 45° relative to the fibril z-axis (Supplementary Fig. 4a). In addition to the different orientation, the peptides show a slightly revised orientation of the amino acid side-chains in each stack (Supplementary Fig. 4b). These data indicate the significant structural plasticity of PNF-18 amyloid fibrils – in particular across the hydrophobic stack interface.

### Molecular basis of PNF-18 fibril polymorphism

In the next step, we compared the structure of Morphology II with the 3D maps obtained with Morphologies I and III. These maps reached spatial resolutions of only 7 to 9 Å (Supplementary Fig. 5) and lacked information about the molecular fibril details. However, we noted that the fibril cross-sections of Morphologies I and III show a number of similarities to Morphology II. All fibril cross sections are roughly rectangular with dimensions of approximately 3 × 4 nm (Morphology I), 3 × 6 nm (Morphology II) and 3 × 8 nm (Morphology III). The three fibrils consist of similarly shaped, elongated density regions that extend side-by-side parallel to the 3 nm side of the fibril cross-section. Morphology II shows six such density regions, representing the six peptides of the fibril cross-section. Morphology I contains four density regions and Morphology III at least eight (Supplementary Fig. 5). Hence, Morphology I seems to contain four peptides in each molecular layer and Morphology III eight. The general structural similarity of the two fibrils to Morphology II implies the existence of common hierarchical structural principles underlying the formation of differently structured PNF-18 amyloid fibrils and that the three fibril morphologies differ mainly in the number of their constituting PFs.

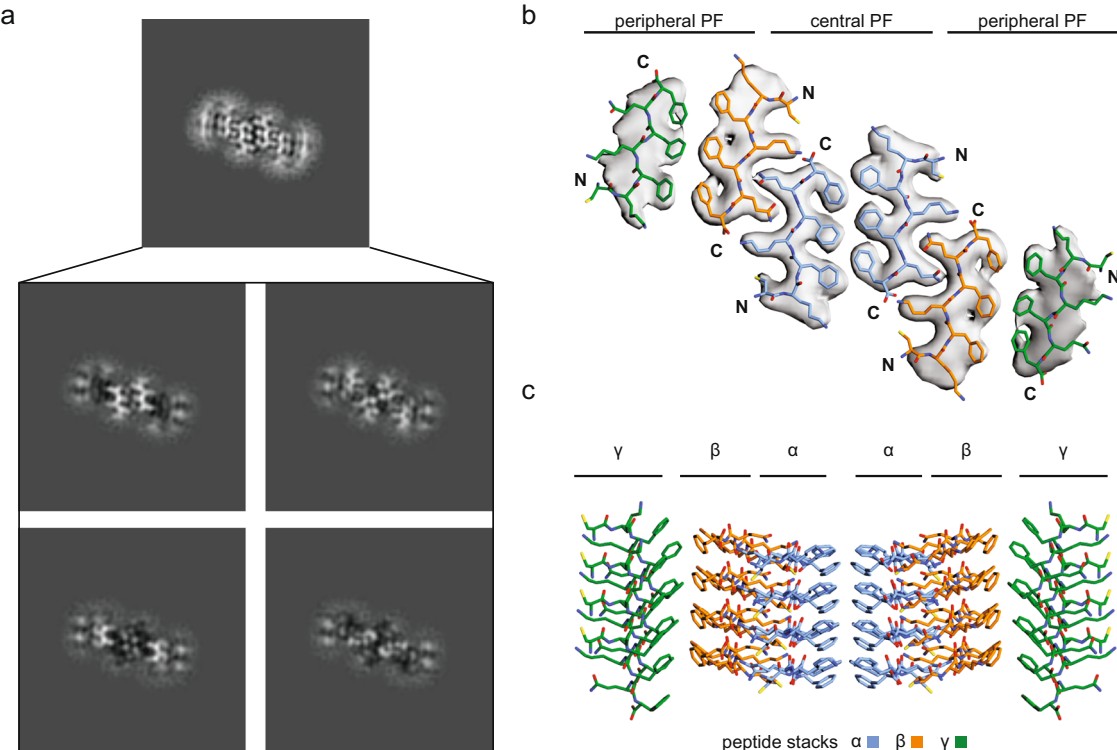

**Fig. 3 | Cryo-EM structure of Morphology II of the in vitro PNF-18 fibrils. a** 4.8 Å tick slice of one molecular layer of the fibril (top). Below are 1.2 Å thick slices through one layer of the reconstructed density of Morphology II (bottom). **b** Cross-sectional view of one molecular layer of the reconstructed density (gray), super-imposed with the molecular model. **c** Side view of a stack of four layers of the molecular model showing the staggering of the peptide stack towards each other.

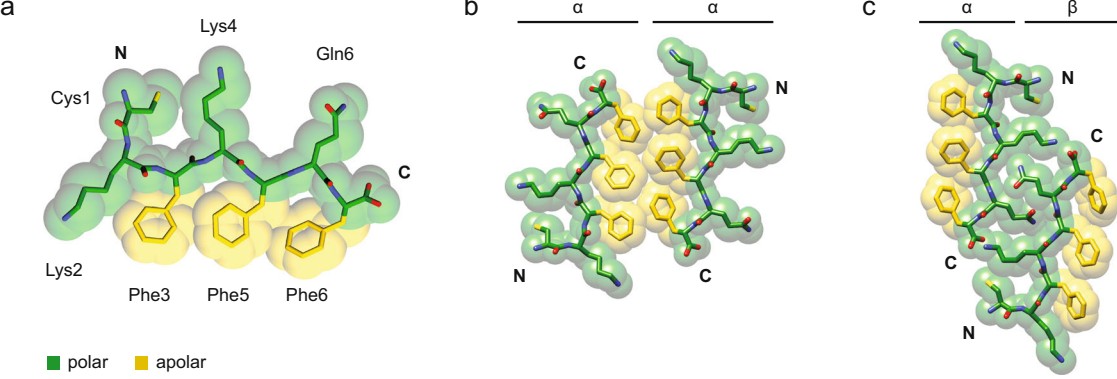

**Fig. 4 | Amphiphilic pattern of the PNF-18 peptide. a** Depiction of the amphiphilic nature of the peptide, showing the polar (green) and apolar (yellow) sites in the molecule. **b** Hydrophobic contact mode between two type α peptide stacks (**c**) Hydrophilic contact mode between a α and a β peptide stack.

## Surface texture of PNF-18 fibrils

Morphology II shows a strongly cationic Coulomb surface potential, which originates mainly from the side chains of Lys2 and Lys4, and the α-amino group of Cys1 (Fig. 5). The only full anionic charge on the surface arises from the α-carboxy groups at the peptide C-terminus of stack γ (Fig. 5a). In the other two peptide stacks (α and β), the C-terminal α-carboxy groups become compensated by the ε-amino groups of Lys4 in the juxtaposed PF (Fig. 5a), as described above for the polar contact mode (Fig. 4c). Due to the similarity of the three fibril morphologies, it is likely that Morphologies I and III show a similar cationic Coulomb surface potential and present similar interactions with viral particles and cellular surfaces as Morphology II. This conclusion is supported by the fact that the virus enhancing activity is almost the same with differently aged solutions of PNF-18 peptide

(Supplementary Fig. 1), although TEM shows clear differences in the fibril morphology (Figs. 1, 2).

The enhancement of viral infectivity by amyloid fibrils is thought to occur via two mechanistic steps: the binding of the fibrils to the viral particles and the binding of the formed virus-fibril complexes to cellular surfaces[15], which can be facilitated by extracellular protrusions[16]. The fibril surface charge will obviously be crucial for mediating these reactions, as was shown previously for non-amyloid transduction enhancers, such as Polybrene[19–21], and dextrans[22–24], as well as (α-helical) Vectofusin®-1[25,26], which were found to be strongly cationic, similar to PNF-18 fibrils. PNF-18 fibrils correspond in these properties to the initially identified amyloid-based enhancers of viral infectivity, which showed positive zeta-potentials[10,12], and their activity could be counteracted by addition of polyanions, such as heparin or dextran

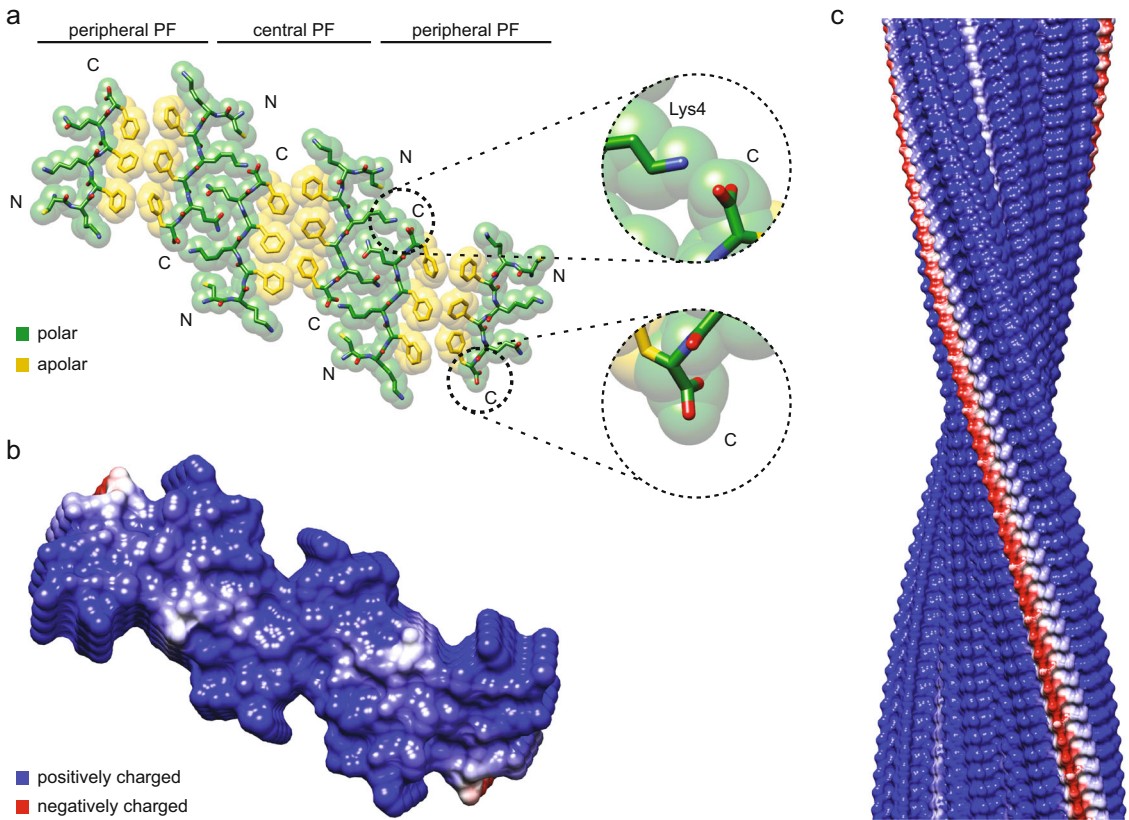

**Fig. 5 | Arrangement and Coulomb potential of Morphology II of the PNF-18 fibrils. a** Cross-sectional view of the molecular model of Morphology II showing the polar (green) and apolar (yellow) sites in the molecule. **b** Coulomb surface potential of the cross-sectional view of Morphology II and (**c**) side view of the fibril.

sulfate, or by replacement of cationic residues with alanine[15]. Anionic amyloid fibrils, such as the ones formed by the peptide AL1, do not enhance the viral infectivity although they are able to bind to viral particles[27]. These data suggested that cationic properties are in particular important for mediating the attachment of the virus-fibril complexes to cellular surfaces. It is interesting to note in this context that amyloid fibrils are known for decades to bind strongly to glycosaminoglycans, which are negatively charged and define the extracellular matrix and surface of many mammalian cells[28].

## Discussion

Amyloid fibrils recently emerged as powerful tools to improve retroviral transduction efficiencies in vitro[6] if not in vivo[8]. Our structure of the virus enhancing fibril formed by peptide PNF-18 provides a molecular view of one such agent and enables a better chemical understanding of the properties of these transduction enhancers as well as of the structural basis of their biological activity. Our structure reveals, in particular, how the cationic chemical groups within PNF-18 become arranged by the peptide conformation and the assembly of the fibril on the surface of the formed assemblies. It shows how the fibril structure helps to mask most of the anionic charges of the α-carboxy groups of residue Phe7 with the ε-amino groups of Lys4 in the polar contact mode (Fig. 5a). Furthermore, our data indicate how the cationic surface properties are maintained irrespective of the specific fibril morphology and why samples containing differently structured PNF-18 fibril morphologies lead to similar transduction efficiencies with HIV-1 particles in our in vitro assay (Supplementary Fig. 1).

In addition to these biological conclusions, the now available cryo-EM data have ramifications for our understanding of the structure of amyloid fibrils and their formation. The data highlight the remarkable structural plasticity of these fibrils in particular when they are formed

from short peptides. The current understanding of peptide cross-β structures is largely dominated by the crystallographic analysis of peptide-derived 3D microcrystals[18,29]. These analyses showed a remarkable persistence of the peptide structure throughout the crystal structure that shows only small, if any, adaptations of the peptide conformation to the specific position of the peptide in the assembly. As the cross-β sheets in the microcrystals were untwisted, these crystal structures differ profoundly from the situation in amyloid fibrils, which are typically twisted assemblies[2]. The twist produces different torsional forces at different positions in the fibril structure and creates different environments for the peptides, depending on their exact radial and azimuthal position. The now available structures of amyloid fibrils from short peptides demonstrate small structural rearrangements in the peptide conformation – specifically within the side chains (Supplementary Fig. 4b)−and in their overall orientation at outer radial positions (Supplementary Fig. 4a). A increased tilt of the peptide molecules at outer radial positions was recently observed in a catalytically active peptide fibril[30], suggesting that the altered tilt helps to compensate the higher torsional strain at outer radial positions.

We noted that the morphology of PNF-18 fibrils is not only dependent on the chemical and physical environment of fibril formation but also on the incubation time. While samples of PNF-18 peptide, which were incubated for only 1 day, contained predominantly thin fibril morphologies, we noted that the samples became progressively more heterogenous as we increased to incubation time and showed a clear trend towards thicker fibril structures (Fig. 1). For example, Morphology I accounted for 63% of the fibrils after day 1 and only 6% of the fibrils after 14 days (Fig. 2c). These data indicate that thin fibrils are kinetically more favorable and represent the first fibrils structures that accumulate during fibril formation. By contrast, thicker amyloid fibril structures are more thermodynamically stable and become more dominating after later time points. Our observations resemble the

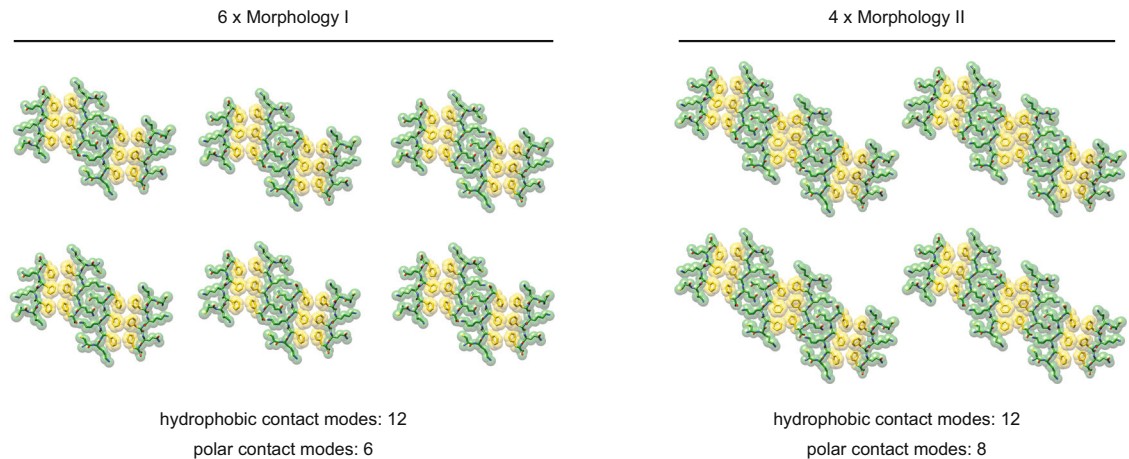

**Fig. 6 | Energetic benefit of the formation of broader fibrils.** Biochemical calculation example of the benefit of the conversion of six fibrils of Morphology I (left) into four fibrils of Morphology II (right). The gain is the formation of two additional polar contact modes.

known maturation of Aβ peptide (and other) amyloid fibrils from thin, metastable protofibrils into thick, mature amyloid fibrils[31].

The structural features of Morphology I, which consists of two PFs and dominates on day 1 (Fig. 2c), and Morphology II, which consists of three PFs and dominates on day 14 (Fig. 2c) implies that the gain in thermodynamic stability on day 14 originates from the additional interactions in the broader fibril. A simple biochemical calculation supports this view: assume that six fibrils of Morphology I convert into four fibrils of Morphology II! Six fibrils of Morphology I show twelve sheet-sheet contacts through the hydrophobic contact mode and six of the polar type (Fig. 6). In case of the four fibrils of Morphology II, there are twelve sheet-sheet contacts through the hydrophobic contact mode and eight of the polar type. That is, the benefit of polymorphic maturation arises, in this simplistic example, from the gain of two polar contact modes.

Although this result is derived here only for the specific case of a small peptide fibril system, similar but substantially more complex calculations may be performed for larger protein systems which consider numerous other factors that may contribute to defining the thermodynamic stability of a fibril structure as well. Yet, polymorphic maturation is an evidently important factor that influences the spectrum of fibril structures that is present in a sample of amyloid fibrils formed in vitro. Whether polymorphic maturation occurs also with the disease-associated amyloid fibrils that are deposited inside the tissue of a patient or animal remains to be established, as biological mechanisms may exist that control the fibril morphological ensemble and its ability to resist cellular fibril degradation and clearance mechanisms[32].

## Methods
### Formation of PNF-18 fibrils
Chemically synthetic PNF-18 peptide was obtained from the group of Prof. Dr. Tanja Weil (Max Planck Institute for Polymer Research, Mainz, Germany). The peptide was incubated at 0.3 mg/mL concentration in 50 mM HEPES buffer, pH 7.0, for a period of up to 60 days at room temperature. After different time points aliquots were withdrawn from the reaction and analyzed as indicated in each experiment.

### TEM analysis of negatively stained samples
For TEM specimen preparation, 4 μL of the sample solution were placed onto a glow-discharged carbon coated, formvar 200 mesh copper grids (C-flat). The sample was incubated on the grid for 1 min at room temperature. Surplus solution was removed with filter paper (Whatman). The grid was washed three times with 10 μL water and

stained three times with 10 μL 2% (w/v) uranyl acetate in water. The dried grids were examined in a JEM-1400 TEM (JEOL) equipped with a F216 camera (TVIPS) that was operated at 120 kV.

### Cryo-EM
C-flat holey carbon grids (CF 1.2/1.3–2 C, Electron Microscopy Sciences) were glow-discharged for 40 s at 20 mA using a PELCO easiGlow glow discharge cleaning system (TED PELLA). 4 μL of sample were then applied on the grid and incubated for 30 s, followed by both side blotting for 8 s (temperature 21°, humidity >90%) and plunging into liquid ethane using a Vitrobot Mark 3 (Thermo Fisher Scientific). Grids were screened using a JEM-2100 transmission electron microscope (Jeol) at 200 kV. High-resolution data sets of the fibrils were recorded with a Titan Krios (Thermo Fisher Scientific) microscope that was equipped with a K2-Summit detector (Gatan) and operated at an acceleration voltage of 300 kV. The width and crossover distance of morphologies were determined from cryo-TEM images using ImageJ software.

### Reconstruction of the 3D map and model building
The correction of the movie frames for the gain reference was done with IMOD[33]. Motion correction and dose-weighting were done using MOTIONCOR 2.1[34]. The contrast transfer function from the motion-corrected images was estimated by using Gctf[35]. Helical reconstruction was carried out by using RELION 3.1[36]. The fibrils were picked manually, and segments were extracted by applying the parameters listed in Supplementary Table 1. A featureless cylinder, which was created by using relion_helix_toolbox, was used as initial 3D model. The resulting reconstructions showed partially separated β-strands in the x–y plane and along the fibril axis that indicated the presence of six identical protein stacks, related by a C2 symmetry. Based on an initial low resolution density map an atomistic model was created and converted to a map by using the molmap command in Chimera[37] and then used as an initial model. Together with the imposing of a C2 symmetry during reconstruction yielded clearly separated β-strands. To further improve the model, 3D classification with local optimization of helical twist and rise and a C2 symmetry were used. The best 3D class was finally reconstructed with local optimization of the helical parameters using 3D auto-refinement. All 3D classification and auto-refine processes were carried out by using a central part of 10% of the intermediate asymmetrical reconstruction. The final reconstruction was post-processed with a soft-edge mask and B-factor sharpened. The estimate of the resolution of the reconstruction was carried out based on the FSC at 0.143 between two independently refined half-maps.

The model was built de novo using the program Coot[38]. A poly-L-Ala chain was traced along the main chain density that was afterward mutated according to residues of the peptide sequence. The structure was then manually refined further in Coot. MolProbity[39] and the comprehensive validation tool in Phenix[40] were used for the analyzation of the atomic clashes, rotamer and Ramachandran outliers and model geometry and the generation of the validation output. Once a satisfactory fit of the main and side-chain density was achieved for one polypeptide chain, a four layered fibril stack comprising 24 peptide molecules was assembled using the pdbsymm tool implemented in Situs[41]. The described process of iterative refinement and modeling was repeated for the fibril stack over and over again, until the refinement converged to produce reasonable density to model fit. The structural statistics for refinement and model building are listed in Supplementary Table 2.

### Sample statistics
The error bars shown in this manuscript report the standard deviation.

### Reporting summary
Further information on research design is available in the Nature Portfolio Reporting Summary linked to this article.

## Data availability
The reconstructed cryo-EM map has been deposited in the Electron Microscopy Data Bank with the accession codes EMD-16930. The coordinates of the fitted atomic model have been deposited in the Protein Data Bank (PDB) under the accession code PDB 8OKR. Source data are provided with this paper.

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

## Acknowledgements
This work was funded by the Deutsche Forschungsgemeinschaft (CRC 1279/2 project A03 to M.F. and J.M.). The authors thank Tanja Weil (Max Planck Institute for Polymer Research, Mainz) for the gift of PNF-18 peptide and for critically reading the manuscript. The authors further acknowledge technical support from Felix Weis (European Molecular Biology Laboratory, Heidelberg), where the 300 kV cryo-EM data were collected.

## Author contributions
T.H., D.S., and Y.-J.L. carried out experiments. T.H., D.S., Y.-J.L., J.M., M.S., and M.F. analyzed the data. M.F. designed research. T.H. and M.F. wrote the paper with comments from all other authors.

## Funding

## Competing interests
The authors declare no competing interests.
