## [Peer Review File · Nature Communications]

REVIEWER COMMENTS

Reviewer #1 (Remarks to the Author):

Overall, I find the paper is well-written and presents novel findings. The authors have used cryo-EM to determine the structures of a retroviral transduction enhancer amyloid produced by a 7-mer peptide PNF-18. The authors have quantified the population of the three major morphologies of PNF-18 at different time points of maturation. Thinner morphologies produced at earlier time points further assembling into thermodynamically stable thicker morphologies upon maturation is an interesting observation. Incubation time as a factor influencing amyloid polymorphism is an important parameter to look at. The mechanism of polymorphic maturation and enhancement of viral transduction proposed by the authors seems plausible.

Major corrections

- Figure 1a: Evidence from NS-TEM analysis for the time dependence of the fibril polymorphism could be consolidated by observing at multiple areas of the grid. How homogeneously or heterogeneously the filaments were spread across the grid should be clearly mentioned. The statistics should take into consideration fibrils found in regions that represent different areas of the grid. A low mag view of the grid (say 500X magnification) or 3000X magnification views of multiple grid squares could be added to the supplementary data.
- Figure 2c: The detailed methodology of how fibrils were chosen for morphology analysis should be mentioned somewhere in the text. From the two datasets collected, a comparison of a number of particles belonging to each Cryo-EM 2D class corresponding to each morphology would give an overall picture of the sample.
- Sheena Radford has been presenting similar observations on fibril maturation in a different system. If that paper is out already, it would be good to clearly link the stories to consolidate the generality of the findings.

Minor corrections

- Line 132: Authors' reported resolution of 2.8 Å does not match well with the maps presented in Figure 3c and Supplementary Figure 3, Resolution calculated from deposited half maps mentioned in the PDB validation document (i.e. 3.40 Å) should be used rather.
- Supplementary Figure 2a b: Labelling for the graphs are missing FSC 0.143 threshold line is missing, 2c width of the fibrils 2D classes should be added, the Fourier peaks should be labelled

Reviewer #2 (Remarks to the Author):

Heerde et al describe the cryo-EM structure of a fibril formed by the peptide PNF-18, which was developed as an enhancer for viral infectivity used in retroviral gene transfer. The development of this peptide and its properties have been described before (Sieste 2021). Designing amyloids for biotechnological applications, in this case using amyloids as enhancers of viral infectivity in gene therapy, is a highly interesting and timely topic.

The authors observe mainly three morphologies of PNF-18, but only morphology 2 could be solved to high-resolution. However the other two morphologies seem to be composed of the same building blocks as morphology 2, so atomic models could be deduced, which seems plausible.

The authors could reveal the building principle of the different morphologies (a polar and a apolar interaction interaction mode between the peptides). It was also observed that the population of the morphologies changes over the course of several days. Morphology 1 apparently is kinetically favored, but higher order assemblies are thermodynamically favored. Interestingly, the change in morphology does not have a strong effect on viral infectivity enhancement. The presented structure, in particular the electrostatic properties of the fibril, contributes to the understanding of the mechanism of how such amyloids enhance viral infectivity and is therefore relevant and interesting.

The cryo-EM structural determination seems well done. The paper is well written and easy to read.

Minor issues:

- Fig. 3a: it would be good to include one image showing the projection along the z-axis over one entire layer (4.75Å). This would give a clearer view of the peptide density.

Typos:

page 8: "... surface potential, which is originates..."

page 9: "The enhancement of viral infectivity by amyloid fibrils is thought *to* occur via two mechanistic steps: "

REVIEWER COMMENTS

Reviewer #1 (Remarks to the Author):

Overall, I find the paper is well-written and presents novel findings. The authors have used cryo-EM to determine the structures of a retroviral transduction enhancer amyloid produced by a 7-mer peptide PNF-18. The authors have quantified the population of the three major morphologies of PNF-18 at different time points of maturation. Thinner morphologies produced at earlier time points further assembling into thermodynamically stable thicker morphologies upon maturation is an interesting observation. Incubation time as a factor influencing amyloid polymorphism is an important parameter to look at. The mechanism of polymorphic maturation and enhancement of viral transduction proposed by the authors seems plausible.

Major corrections

- Figure 1a: Evidence from NS-TEM analysis for the time dependence of the fibril polymorphism could be consolidated by observing at multiple areas of the grid. How homogeneously or heterogeneously the filaments were spread across the grid should be clearly mentioned. The statistics should take into consideration fibrils found in regions that represent different areas of the grid. A low mag view of the grid (say 500X magnification) or 3000X magnification views of multiple grid squares could be added to the supplementary data.

Response:

We truly appreciate the thoughtful comments and feedback of this referee. While we could add a 500x micrograph for each sample, this micrograph would originate from only one grid window. Our measurements are based on multiple 20,000x micrographs (at least 5 per sample) that originate from different parts of the grid. These are now shown in the Source Data file.

- Figure 2c: The detailed methodology of how fibrils were chosen for morphology analysis should be mentioned somewhere in the text. From the two datasets collected, a comparison of a number of particles belonging to each Cryo-EM 2D class corresponding to each morphology would give an overall picture of the sample.

Response:

For the morphological analysis the fibrils were classified based on their width and cross-over distance. Therefore, 500 fibrils per sample were measured and based on this classified as Morphology I to III or others. These values were used to calculate the relative abundance of each morphology. To clarify this, we added "The relative abundance of each morphology was determined based on the cross-over distance and the width of 500 randomly chosen fibrils from each sample, [...]." to the text. The quantification of the abundance based on the particle number or 2D classes is in our case not very meaningful, as the fibril morphologies were picked separately and their numbers and lengths do not necessarily reflect the true fibril distribution.

- Sheena Radford has been presenting similar observations on fibril maturation in a different system. If that paper is out already, it would be good to clearly link the stories to consolidate the generality of the findings.

Response:

Thank you for your suggestion. We are always happy to cite Sheena Radford as she always publishes well-written manuscripts and exciting research. However, we are not sure which manuscript is exactly meant by the referee.

Minor corrections

- Line 132: Authors' reported resolution of 2.8 Å does not match well with the maps presented in Figure 3c and Supplementary Figure 3, Resolution calculated from deposited half maps mentioned in the PDB validation document (i.e. 3.40 Å) should be used rather.

Response:

We thank this referee for this very constructive comment. The different numbers arise because of the use of masks. The resolution reported by the PDB validation report refers to the unmasked-calculated FSC curve and is compared in this report to the masked FSC that was obtained from

Relion. Indeed, the unmasked FSC curve of the PDB validation report is almost identical to the unmasked FSC curve calculated by Relion (Supplementary Figure 2a). Also the local resolution map was calculated based on the unmasked half-maps. For clarification, we now added "The local resolution was calculated based on the unmasked half-maps." to the figure legend of the local resolution, and we now also mention the unmasked resolution in the Supplementary Table 2.

- Supplementary Figure 2a b: Labelling for the graphs are missing FSC 0.143 threshold line is missing, 2c width of the fibrils 2D classes should be added, the Fourier peaks should be labelled

Response:

Thank you for your suggestion. We added the requested labels as well as a scale bar for the 2D classes.

Reviewer #2 (Remarks to the Author):.

Heerde et al describe the cryo-EM structure of a fibril formed by the peptide PNF-18, which was developed as an enhancer for viral infectivity used in retroviral gene transfer. The development of this peptide and its properties have been described before (Sieste 2021). Designing amyloids for biotechnological applications, in this case using amyloids as enhancers of viral infectivity in gene therapy, is a highly interesting and timely topic.

The authors observe mainly three morphologies of PNF-18, but only morphology 2 could be solved to high-resolution. However the other two morphologies seem to be composed of the same building blocks as morphology 2, so atomic models could be deduced, which seems plausible.

The authors could reveal the building principle of the different morphologies (a polar and a apolar interaction interaction mode between the peptides). It was also observed that the population of the morphologies changes over the course of several days. Morphology 1 apparently is kinetically favored, but higher order assemblies are thermodynamically favored. Interestingly, the change in morphology does not have a strong effect on viral infectivity enhancement. The presented structure, in particular the electrostatic properties of the fibril, contributes to the understanding of the mechanism of how such amyloids enhance viral infectivity and is therefore relevant and interesting.

The cryo-EM structural determination seems well done. The paper is well written and easy to read.

Minor issues:

- Fig. 3a: it would be good to include one image showing the projection along the z-axis over one entire layer (4.75Å). This would give a clearer view of the peptide density.

Response:

We gratefully acknowledge the supportive comments of this referee. As requested, we added a z-projection over one entire layer.

Typos:

page 8: "... surface potential, which is originates..."

page 9: "The enhancement of viral infectivity by amyloid fibrils is thought *to* occur via two mechanistic steps: "

Response:

Thank you, we changed it.

REVIEWERS' COMMENTS

Reviewer #1 (Remarks to the Author):

The authors addressed all my comments